# Range Entropy: A Bridge between Signal Complexity and Self-Similarity

**DOI:** 10.3390/e20120962

**Published:** 2018-12-13

**Authors:** Amir Omidvarnia, Mostefa Mesbah, Mangor Pedersen, Graeme Jackson

**Affiliations:** 1The Florey Institute of Neuroscience and Mental Health, Austin Campus, Heidelberg, VIC 3084, Australia; 2Faculty of Medicine, Dentistry and Health Sciences, The University of Melbourne, VIC 3010, Australia; 3Department of Electrical and Computer Engineering, Sultan Qaboos University, Muscat 123, Oman; 4Department of Neurology, Austin Health, Melbourne, VIC 3084, Australia

**Keywords:** approximate entropy, sample entropy, range entropy, complexity, self-similarity, Hurst exponent

## Abstract

Approximate entropy (*ApEn*) and sample entropy (*SampEn*) are widely used for temporal complexity analysis of real-world phenomena. However, their relationship with the Hurst exponent as a measure of self-similarity is not widely studied. Additionally, *ApEn* and *SampEn* are susceptible to signal amplitude changes. A common practice for addressing this issue is to correct their input signal amplitude by its standard deviation. In this study, we first show, using simulations, that *ApEn* and *SampEn* are related to the Hurst exponent in their tolerance *r* and embedding dimension *m* parameters. We then propose a modification to *ApEn* and *SampEn* called *range entropy* or *RangeEn*. We show that *RangeEn* is more robust to nonstationary signal changes, and it has a more linear relationship with the Hurst exponent, compared to *ApEn* and *SampEn*. *RangeEn* is bounded in the tolerance *r*-plane between 0 (maximum entropy) and 1 (minimum entropy) and it has no need for signal amplitude correction. Finally, we demonstrate the clinical usefulness of signal entropy measures for characterisation of epileptic EEG data as a real-world example.

## 1. Introduction

Complexity is a global concept in data analysis that is observed in a wide range of real-world phenomena and systems including biological signals [1,2,3,4,5,6], brain dynamics [7,8,9], mechanical systems [10,11], climate change [12], volcanic eruption [13], earthquakes [14], and financial markets [15]. It is difficult to provide a formal definition for signal complexity. This concept, however, can be approached as a mid-point situation between signal regularity and randomness. From this perspective, complexity can be defined as the amount of nonlinear information that a time series conveys over time. Highly random fluctuations (such as white noise as an extreme case) have very low complexity, because they present no regular pattern in their dynamical behaviour. Real-world phenomena, on the other hand, usually contain spreading patterns of nonlinear ’structured activity’ across their frequency components and temporal scales. Dynamics of the brain or fluctuation of stock markets are examples of complex processes. Despite the importance of complexity in science, its quantification is not straightforward. Time-frequency distributions and wavelet transforms [16] are examples of analysis tools for capturing signal dynamics, but they may be insensitive to nonlinear changes.

A promising avenue for understanding temporal complexity is through signal entropy analysis, a family of methods rooted in information theory. Entropy rate of a random process is defined as the average rate of generation of new information [17]. In this context, independent and identically distributed white noise is assumed to have maximal entropy and disorder. This is because identically distributed white noise has a normal distribution where each upcoming time point contains new information. On the other hand, a completely periodic signal with a repeating pattern of constant values will lead to minimal entropy, as there is no generation of new information. The most prominent types of signal entropy measures include Shannon entropy [17], Renyi entropy [18], Kolmogorov entropy [19,20], Kolmogorov–Sinai entropy [21], Eckmann–Ruelle entropy [22], approximate entropy (*ApEn*) [23], sample entropy (*SampEn*) [24], and multi-scale entropy [25]. See [26] for more examples of entropy-based signal measures.

Among the aforementioned signal entropy measures, *ApEn* and *SampEn* are two of the most commonly used measures in contemporary science, especially in the analysis of biological signals [27]. Like *ApEn*, *SampEn* resembles a template-matching search throughout the input signal with two main parameters: embedding dimension *m* and tolerance *r*. The former governs the length of each segment (template) to be searched and the later controls the level of similarity between segments. In fact, *SampeEn* stems from *ApEn* after addressing some of its limitations including inconsistency over the parameter *r* and strong dependency to the input signal length [24]. However, both measures still suffer from sensitivity to signal amplitude changes. Another important aspect of these measures is their inverse relationship with the Hurst exponent as a measure of self-similarity in signals [28]. The analysis of this link, however, deserves more attention.

In this study, we investigate the behaviour of *ApEn* and *SampEn* in the presence of self-similarity and examine their relationship with the Hurst exponent through their tolerance and embedding dimension parameters. We also address the issue of sensitivity to signal amplitude changes in *ApEn* and *SampEn* by developing modified versions called *range entropies* or *RangeEn*. We compare *RangeEn* with *ApEn* and *SampEn* from different perspectives using multiple simulations. Finally, we demonstrate the capacity of signal entropy measures for epileptic EEG characterisation. A Python implementation of this study is publicly available at https://github.com/omidvarnia/RangeEn.

## 2. Materials and Methods

### 2.1. Signal Complexity Analysis

#### 2.1.1. Reconstructed Phase Space

Numerical computation of signal entropy for a uniformly-sampled signal $x=\{x_{1},x_{2},\ldots ,x_{N}\}$ can be done through the concept of reconstructed phase space [27]. It represents the dynamical states of a system with state variables $X_{i}^{m,\tau }$ defined as [29]:(1)$$\left\{ \begin{array}{c} X_{i}^{m,\tau }=\left\{x_{i},\;x_{i+\tau },\ldots ,\;x_{i+(m-1)\tau }\right\}\\ i=1,\ldots ,N-(m-1)\tau \end{array},\right.$$
where *m* denotes the embedding dimension and τ is the delay time. $X_{i}^{m,\tau }$ represents a state vector in an *m*-dimensional phase space $V_{x}$. The parameter τ is also referred to as scale.

Given a reconstructed state vector $X_{i}^{m,\tau }$, it is possible to partition $V_{x}$ into small non-overlapping and equisized regions $\varepsilon_{k}$, so that ${⋃}_{k}\varepsilon_{k}=V_{x}$ and ${⋂}_{k}\varepsilon_{k}=0$. Signal entropy can then be computed by assigning a probability value $p_{k}$ to each region as the probability of visiting the phase trajectory [27].

From now on, we consider a special case of $V_{x}$ where τ=1. In this case, the state vector $X_{i}^{m,\tau }$ is reduced to a vector sequence of $x_{i}$ through to $x_{i+m-1}$, i.e.,:(2)$$X_{i}^{m}=\left\{x_{i},\;x_{i+1},\;\ldots \;,\;x_{i+m-1}\right\},\;i=1,...,N-m+1.$$

#### 2.1.2. Approximate Entropy

Let each $X_{i}^{m}$ in Equation (Equation 2) be used as a template to search for *neighbouring* samples in the reconstructed phase space. Two templates $X_{i}^{m}$ and $X_{j}^{m}$ are matching if their relative distance is less than a predefined tolerance *r*. The distance function used in both *ApEn* and *SampEn* is the Chebyshev distance defined as $d_{chebyshev}\left(X_{i}^{m},X_{j}^{m}\right):=\max_{k}\left(\right|x_{i+k}-y_{j+k}\left|,\;k=0,\ldots ,m-1\right)$. It leads to an *r-neighbourhood* conditional probability function $C_{i}^{m}\left(r\right)$ for any vector $X_{i}^{m}$ in the phase space $V_{x}$:(3)$$C_{i}^{m}\left(r\right)=\frac{1}{N-m+1}B_{i}^{m}\left(r\right),\;i=1,\ldots ,N-m+1,$$
where
(4)$$\left\{\begin{array}{c} B_{i}^{m}\left(r\right)=\left\{No.\;of\;X_{j}^{m}s\;|\;d_{chebyshev}\left(X_{i}^{m},X_{j}^{m}\right)\leq \;r\right\}\\ j=1,...,N-m+1\; \end{array}\right..$$

Let $\Phi^{m}\left(r\right)$ be the sum of natural logarithms $C_{i}^{m}\left(r\right)$; that is,
(5)$$\Phi^{m}\left(r\right)=\sum_{i=1}^{N-m+1}ln\;C_{i}^{m}\left(r\right).$$

The rate of change in $\Phi^{m}\left(r\right)$ along the embedding dimension *m* is called the Eckmann–Ruelle entropy and is defined as [22]:(6)$$H_{ER}=\lim_{r\to 0}\lim_{m\to 0}\lim_{N\to \infty }\Phi^{m}\left(r\right)-\Phi^{m+1}\left(r\right).$$

An approximation of $H_{ER}$, proposed by Pincus through fixing *r* and *m* in Equation (Equation 6), is called approximate entropy (*ApEn*) [23,30]:(7)$$ApEn=\lim_{N\to \infty }\Phi^{m}\left(r\right)-\Phi^{m+1}\left(r\right),\;\;\;r,\;m\;fixed.$$

*ApEn* quantifies the mean negative log probability that an *m*-dimensional state vector will repeat itself at dimension (*m* + 1). It is recommended that the tolerance is corrected as r×SD (*SD* being the standard deviation of x) to account for amplitude variations across different signals.

#### 2.1.3. Sample Entropy

As Equations (Equation 3) and (Equation 4) suggest, *ApEn* allows for the self-matching of templates $X_{i}^{m}$ in the definition of $C_{i}^{m}\left(r\right)$ to avoid the occurrence of *ln(0)* in its formulation [30]. However, this will result in an unwanted bias that occurs in particular for short signal lengths (small *N*). Inconsistency of *ApEn* over the tolerance parameter *r* has also been reported [24,30]. In order to address these issues, sample entropy was developed by updating $B_{i}^{m}\left(r\right)$ in Equation (Equation 4) [24]:(8)$$\left\{\begin{array}{c} B_{i}^{m}\left(r\right)=\left\{No.\;of\;X_{j}^{m}s\;|\;d_{chebyshev}\left(X_{i}^{m},X_{j}^{m}\right)\leq \;r\right\}\\ j=1,\ldots ,N-m,\;j\neq i \end{array}\right.,$$
averaging over time as:(9)$$B_{m}^{r}=\frac{1}{N-m}\sum_{i=1}^{N-m}B_{i}^{m}\left(r\right).$$

Sample entropy is then defined as:(10)$$SampEn=\lim_{N\to \infty }-ln\;\frac{B_{m+1}^{r}}{B_{m}^{r}}.$$

There are three major differences between *SampEn* and *ApEn*:(1)Conditional probabilities of *SampEn*, i.e., $B_{i}^{m}\left(r\right)$ in Equation (Equation 8), are obtained without self-matching of the templates $X_{i}^{m}$.(2)Unlike *ApEn*, which takes the logarithm of each individual probability value (see Equation (Equation 5)), *SampEn* considers the logarithm of the sum of probabilities in the phase space (see Equations (Equation 9) and (Equation 10)).(3)*ApEn* is defined under all circumstances due to its self-matching, while *SampEn* can sometimes be undefined, as $B_{m}^{r}$ and $B_{m+1}^{r}$ in Equation (Equation 10) are allowed to be zero.

Since $d_{chebyshev}\left(X_{i}^{m},X_{j}^{m}\right)$ is always smaller than or equal to $d_{chebyshev}\left(X_{i}^{m+1},X_{j}^{m+1}\right)$, $B_{m+1}^{r}$ is less than $B_{m}^{r}$ for all values of *m*. Therefore, *SampEn* is always non-negative [24]. The parameter set of *m* = 2 and *r* between 0.2 to 0.6 has been widely used for extracting *SampEn* in the literature [9,24,28].

### 2.2. Signal Self-Similarity Analysis

#### 2.2.1. Self-Similar Processes

The time series x(t) is self-similar or scale-invariant if it repeats the same statistical characteristics across multiple temporal scales [31]. In this case, scaling along the time axis by a factor of *a* requires a rescaling along the signal amplitude axis by a factor of $a^{H}$; that is, $x\left(at\right)=a^{H}x\left(t\right)$ for all t>0, a>0, and H>0. The Hurst exponent is a common measure of quantifying self-similarity in signals. Intuitively, the more a signal is self-similar, the more its long-term memory increases. Given the definition of signal entropy as ‘the average rate of generation of new information’ [17], we expect a link between signal entropy and self-similarity. This can be investigated by looking into the signal entropy values of time series with certain degrees of self-similarity. Fractional Levy and Brawnian motions (*fLm* and *fBm*, respectively) are well suited for this purpose. The *fBm* signal $B_{H}\left(t\right)$ is a continuous-time Gaussian process whose difference also leads to a Gaussian distribution, and its self-similarity level is controlled by its Hurst parameter. It is given by [31]:(11)$$B_{H}\left(t\right)=\int_{-\infty }^{\infty }\left\{{\left(t-u\right)}_{+}^{H-1/2}-{\left(-u\right)}_{+}^{H-1/2}\right\}\;\;dB\left(u\right),$$
where *H* is the Hurst exponent (0<H<1), ${\left(x\right)}_{+}:=max\left(x,0\right)$. B(t) is an ordinary Brownian motion, a spacial case at H=0.5, whose frequency spectrum follows the $1/f^{2}$ pattern. *fBm* has the following covariance function:(12)$$E\left\{B_{H}\left(t\right)B_{H}\left(s\right)\right\}=\frac{1}{2}\left(t^{2H}+s^{2H}-|t-s{|}^{2H}\right),\;\;t,s\geq 0.$$

It represents self-similarity (or long-term memory) for H>0.5 and anti self-similarity (or short-term memory) for H<0.5. A more general form of *fBm* is the *fLm* which is defined based on α-stable Levy processes $L_{\alpha }\left(t\right)$ with the following *characteristic function* (the Fourier transform of the probability density function) [32]:(13)$$f\left(x\right)=\frac{1}{\pi }\int_{0}^{\infty }e^{-{\left|Ck\right|}^{\alpha }}\;cos\left(kx\right)\;\;dk,$$
where α is the Levy index (0<α≤2), and C>0 is the scale parameter controlling the standard deviation of Gaussian distributions. The *fLm* signal $Z_{H}^{\alpha }\left(t\right)$ is given by [31]:(14)$$Z_{H}^{\alpha }\left(t\right)=\int_{-\infty }^{\infty }\left\{{\left(t-u\right)}_{+}^{d}-{\left(-u\right)}_{+}^{d}\right\}\;\;dL_{\alpha }\left(u\right),$$
where d=H−1/α. For α=2, Equation (Equation 13) is reduced to the characteristic function of a Gaussian distribution and $Z_{H}^{\alpha }\left(t\right)$ is converted to *fBm*.

#### 2.2.2. Rescaled Range Analysis for Self-Similarity Assessment

A commonly used approach for estimating the Hurst exponent of an *N*-long time series x is through rescaled range analysis [33]. It applies a multi-step procedure on $x=\{x_{1},x_{2},\ldots ,x_{N}\}$ as follows:(1)Divide x into *n* equisized non-overlapping segments $x_{n}^{s}$ with the length of N/n, where s=1,2,3,…,n and n=1,2,4,…. This process is repeated as long as $x_{n}^{s}$ has more than four data points.(2)For each segment $x_{n}^{s}$,
(a)Center it as $y_{n}^{s}=x_{n}^{s}-m_{n}^{s}$, where $m_{n}^{s}$ is the mean of $x_{n}^{s}$. $y_{n}^{s}$ shows the deviation of $x_{n}^{s}$ from its mean.(b)Compute the cumulative sum of centered segment $y_{n}^{s}$ as $z_{n}^{s}=\sum_{i=1}^{N/n}y_{n}^{s}\left(i\right)$. $z_{n}^{s}$ shows the total sum of $y_{n}^{s}$ as it proceeds in time.(c)Calculate the largest difference within the cumulative sum $z_{n}^{s}$, namely,
(15)$$R_{n}^{s}=\max_{k}z_{n}^{s}\left(k\right)-\min_{k}z_{n}^{s}\left(k\right).$$(d)Calculate the standard deviation of $x_{n}^{s}$ as $S_{n}^{s}$ and obtain its rescaledrange as $R_{n}^{s}/S_{n}^{s}$.(3)Compute the average rescaled range at *n* as $R\left(n\right)/S\left(n\right)=\left(1/n\right)\sum_{s=1}^{n}R_{n}^{s}/S_{n}^{s}$.

The average rescaled range R(n)/S(n) is modelled as a power law function over *n* whose asymptotic behaviour represents the Hurst exponent:(16)$$\lim_{n\to \infty }E\left\{R\left(n\right)/S\left(n\right)\right\}=Cn^{H}.$$

*H* can be estimated as the slope of the logarithmic plot of the rescaled ranges versus *ln(n)*. The main idea behind rescaled range analysis is to quantify the fluctuations of a signal around its stable mean [33]. Next, we relate *ApEn* and *SampEn* with rescaled range analysis through their embedding dimension parameter *m*. We then introduce a change into these measures, which makes them more sensitive to self-similarity of signals. We will show the link between entropy measures and the Hurst exponent in Section 3 through simulations of the *fBm* and *fLm* processes.

### 2.3. Complexity and Self-Similarity Analyses Combined

#### 2.3.1. *RangeEn*: A Proposed Modification to *ApEn* and *SampEn*

Both *ApEn* and *SampEn* aim to extract the conditional probabilities of $B_{i}^{m}\left(r\right)$ by computing the Chebyshev distance between two templates (or state vectors) $X_{i}^{m}$ and $X_{j}^{m}$ in the reconstructed *m*-dimensional phase space, as shown in Equations (Equation 4) and (Equation 8). The idea here is to estimate *the (logarithmic) likelihood that runs of patterns that are close remain close on next incremental comparisons* [30]. The closer the two states stay together in the reconstructed phase space over time, the less *change* they will introduce into the signal dynamics. The idea of quantifying the Chebyshev distance between two state vectors originated from the seminal paper by Takens [34].

Although $d_{chebyshev}\left(X_{i}^{m},X_{j}^{m}\right)$ can provide useful information about the variation of state vectors, it has two limitations. First, it is not normalised as it has no upper limit. It leads to an unbounded range for the tolerance parameter *r* in the conditional probabilities $B_{i}^{m}\left(r\right)$ (see Equations (Equation 4) and (Equation 8)). Second, it only considers the maximum element-wise difference between two state vectors, so it is blind to the lower limit of this differences. To address these issues, we adapt the general idea behind the average rescaled range R(n)/S(n) in Equation (Equation 16) and propose an updated version of distance function for *ApEn* and *SampEn* as follows:(17)$$\left\{\begin{array}{c} d_{range}\left(X_{i}^{m},X_{j}^{m}\right)\;=\frac{\max_{k}|x_{i+k}-x_{j+k}|\;-\;\min_{k}|x_{i+k}-x_{j+k}|}{\max_{k}|x_{i+k}-x_{j+k}|\;+\;\min_{k}|x_{i+k}-x_{j+k}|}\\ k=0,\ldots ,m-1 \end{array}\right..$$

In the spacial case of a two-dimensional reconstructed phase space (*m* = 2), $d_{range}\left(X_{i}^{m},X_{j}^{m}\right)$ is reduced to the simple form of (a−b)/(a+b), where $a=\max\{\left(x_{i}-x_{j}\right),\left(x_{i+1}-x_{j+1}\right)\}$ and $b=\min\{\left(x_{i}-x_{j}\right),\left(x_{i+1}-x_{j+1}\right)\}$. In fact, $d_{range}$ considers the stretching of state vectors across time and dimension. In contrast to $d_{chebyshev}\left(X_{i}^{m},X_{j}^{m}\right)$, the proposed $d_{range}\left(X_{i}^{m},X_{j}^{m}\right)$ is normalised between 0 and 1. It also recognises the range of element-wise differences between $X_{i}^{m}$ and $X_{i}^{m}$ by combining the absolute value, *min* and *max* operators. $d_{range}\left(X_{i}^{m},X_{j}^{m}\right)$ is defined over all values, except for identical *m*-dimensional segments where the denominator in Equation (Equation 17) becomes zero.

Strictly speaking, $d_{range}\left(X_{i}^{m},X_{j}^{m}\right)$ is not a distance *per se*, because it does not satisfy all conditions of a distance function. For any two equilength vectors $v_{1}$ and $v_{2}$, these requirements are defined as follows [35]:(18)$$\left\{\begin{array}{ll} (1)\;dist\left(v_{1},v_{2}\right)\geq 0 & \left(non-negativity\right)\\ (2)\;dist\left(v_{1},v_{2}\right)=dist\left(v_{2},v_{1}\right) & \left(symmetry\right)\\ (3)\;dist\left(v_{1},v_{1}\right)=0 & \left(reflexivity\right) \end{array}\right..$$

$d_{range}\left(X_{i}^{m},X_{j}^{m}\right)$ violates the first and third conditions, as it is undefined for equal templates. In fact, $d_{range}\left(X_{i}^{m},X_{j}^{m}\right)$ does not necessarily increase as $X_{i}^{m}$ and $X_{j}^{m}$ become farther away from one another by other definitions of distance functions. For instance, assume all elements of $X_{i}^{m}$ are increased by a constant positive value. The numerator of $d_{range}$ then remains unchanged, while the denominator increases, leading to a reduction in $d_{range}$, even though the Euclidean distance between $X_{i}^{m}$ and $X_{j}^{m}$ has increased. Having this in mind, we have referred to $d_{range}\left(X_{i}^{m},X_{j}^{m}\right)$ as a distance function throughout this paper for practicality. By replacing $d_{chebyshev}\left(X_{i}^{m},X_{j}^{m}\right)$ in Equations (Equation 4) and (Equation 8) with $d_{range}\left(X_{i}^{m},X_{j}^{m}\right)$, we update *ApEn* and *SampEn* as two new range entropy measures, i.e., $RangeEn_{A}$ and $RangeEn_{B}$, respectively.

#### 2.3.2. Properties of *RangeEn*

*Property 1: RangeEn is more robust to nonstationary amplitude changes*. Unlike *SampEn* and *ApEn*, which are highly sensitive to signal amplitude changes, *RangeEn* is less affected by variation in the magnitude of signals. This originates from the in-built normalisation step in $d_{range}\left(X_{i}^{m},X_{j}^{m}\right)$, which is directly applied to the amplitude of all templates.

*Property 2: In terms of r, RangeEn is constrained in the interval [0,1]*. It becomes more obvious if we rewrite Equation (Equation 4) (and similarly, Equation (Equation 8)) as:(19)$$B_{i}^{m}\left(r\right)=\sum_{j=1}^{N-m+1}\Psi \left(r-d_{range}\left(X_{i}^{m},X_{j}^{m}\right)\right),$$
where Ψ(.) is the Heaviside function defined as:(20)$$\Psi \left(a\right)=\left\{\begin{array}{cc} 0 & a<0\\ 1 & a\geq 0 \end{array}\right..$$

Since $d_{range}\left(X_{i}^{m},X_{j}^{m}\right)$ is normalised, we conclude from Equation (Equation 19) that:(21)$$\left\{\begin{array}{c} RangeEn_{A}:\;B_{i}^{m}\left(r\right)=N-m+1\;\;\forall \;r\geq 1\\ RangeEn_{B}:\;B_{i}^{m}\left(r\right)=N-m\;\;\;\;\;\;\;\forall \;r\geq 1 \end{array}\right..$$

This ensures that both conditional probability functions $C_{i}^{m}\left(r\right)$ in Equation (Equation 3) and $B_{m}^{r}$ in Equation (Equation 9) will always be equal to 1 for r≥1 leading to the following property for $RangeEn_{A}$ and $RangeEn_{B}$:(22)$$\left\{\begin{array}{cc} RangeEn_{A}\left(x,m,r\right)=0 & \forall \;r\geq 1\\ RangeEn_{B}\left(x,m,r\right)=0 & \forall \;r\geq 1 \end{array}\right..$$

*Property 3: RangeEn is more sensitive to the Hurst exponent changes*. We will show through simulations that all of *ApEn*, *SampEn*, and *RangeEn* reflect self-similarity properties of the signals in the *r* and *m* domains to different extents. However, *ApEn* and *SampEn* may become insensitive to self-similarity over a significant interval of Hurst exponents, while *RangeEn* still preserves its link.

### 2.4. Simulations

We used simulated data in order to test the behaviour of *ApEn*, *SampEn*, and *RangeEn* on random processes.

#### 2.4.1. Synthetic Data

We simulated 100 realisations of Gaussian white noise (*N(0,1)*), pink (1/f) noise, and brown noise ($1/f^{2}$) and extracted their different entropy estimates across a range of signal lengths and tolerance parameters *r*. We used Python’s *acoustics* library (https://pypi.org/project/acoustics/) to generate noise signals.

We also generated a range of fixed-length *fBm* and *fLm* signals (*N* = 1000) with pre-defined Hurst exponents ranging from 0.01 (minimal self-similarity) to 0.99 (maximal self-similarity) with the increasing step of ΔH = 0.01. We fixed the α parameter of all fractional Levy motions to 1. We used Python’s *nolds* library (https://pypi.org/project/nolds/) to simulate the *fBm* time series and *flm* (https://github.com/cpgr/flm) library to generate *fLm* signals.

#### 2.4.2. Tolerance Parameter *r* of Entropy and the Hurst Exponent

For each of the *fBm* and *fLm* signals at different self-similarity levels, we set the embedding dimension parameter *m* to 2 (a widely used value across the literature) and computed different entropies over a span of tolerance values *r* from 0.01 to 1 with 0.01 increasing steps. In this way, we investigated the relationship between a systematic increase in self-similarity (modelled by the Hurst exponent) and the tolerance parameter *r* in the measures. For each *r*-trajectory, we estimated the slope of a fitted line to the entropy measures with respect to *log(r)* and called this quantity *r-exponent*.

#### 2.4.3. Embedding Dimension *m* of Entropy and the Hurst Exponent

Similar to Section 2.4.2, this time we fixed the tolerance parameter *r* to 0.2 (a common choice in previous studies) and investigated entropy measures over different embedding dimensions *m* from 2 to 10. Therefore, we examined the relationship between a systematic change in the Hurst exponent and the embedding dimension *m*. For each *m*-trajectory, we estimated the slope of a fitted line to the entropy measures with respect to *log(m)* and called this quantity *m-exponent*.

For both analyses described in Section 2.4.2 and Section 2.4.3, we did not perform line fitting for those time series whose extracted entropy measures were undefined for at least one *r* or *m* values. We repeated the above tests with and without amplitude correction (i.e., dividing the signal amplitude by its standard deviation). This correction step is recommended for *ApEn* and *SampEn* analyses, as it can reduce their sensitivity to differences in signal amplitudes (see [24]).

### 2.5. Epileptic EEG Datasets

We used three out of five datasets of a public epileptic EEG database [36] in our study. Each dataset consisted of 100 single-channel EEG segments with the length of 23.6 s and sampling frequency of 173.61 Hz (*N* = 4097) which were randomised over recording contacts and subjects. Datasets *C*, *D*, and *E* of [36] are intracranial EEG (iEEG) from five epilepsy patients who had epilepsy surgery at the hippocampal area and became seizure-free after that. Dataset *C* was recorded from the hippocampal formation on the opposite (contralateral) side of seizure focus, while dataset *D* was recorded from the hippocampal area on the seizure (ipsilateral) side. Both datasets *C* and *D* were obtained during interictal (seizure-free) intervals. In contrast, dataset *E* covered ictal (seizure) intervals only.

All datasets were obtained using a 128-channel EEG recorder with common average referencing. Additionally, eye movement artefacts and strong pathological activities were identified and removed from the signals through visual inspection. A band-pass filter of 0.53–40 Hz was applied to the data. See [36] for more details about these datasets.

## 3. Results

### 3.1. Sensitivity to Signal Length

We simulated three color noise types at different lengths varying from 50 to 1000 samples increasing with 10-sample increasing steps. One hundred realisations of each noise type were generated. Four entropy measures (*ApEn*, *SampEn*, $RangeEn_{A}$, and $RangeEn_{B}$) were then computed from the simulated noise signals. For all entropy measures, we fixed the dimension *m* to 2 and the tolerance *r* to 0.2. Figure 1 illustrates the errorbar plot of each entropy measure for the three noise types over different signal lengths. As the figure suggests, the variations of $RangeEn_{A}$ and $RangeEn_{B}$ are smaller than both *ApEn* and *SampEn* over different lengths. Among the four measures, *SampEn* has the largest standard deviations (poor repeatability) at each signal length, especially for shorter signals. A common observation in all measures is that their standard deviation increases and their mean decreases by increasing the exponential decay in the frequency domain, given a higher spectral exponent of Brown noise compared to pink noise and of pink noise compared to white noise. *ApEn* is the most sensitive measure to signal length, as its mean tends to change (almost linearly) with the data length. *RangeEn* measures present a more stable mean (in contrast to *ApEn*) with small variance (in contrast to *SampEn*).

### 3.2. The Role of Tolerance *r*

To investigate the effect of tolerance *r* on entropy measures, we again simulated three noise types at the fixed length of *N* = 1000 samples. We computed the measures at *m* = 2, but over a range of tolerance values *r* from 0.01 to 1 in increments of 0.01. Figure 2 illustrates the entropy patterns in the *r*-plane for each noise type. Five observations can be drawn from this analysis. First, both $RangeEn_{A}$ and $RangeEn_{B}$ reach zero at *r* = 1. This is not the case for *ApEn* and *SampEn*. Second, *SampEn* shows the highest standard deviation, in particular at low *r* values (r≤0.3). Third, $RangeEn_{A}$ has the highest number of undefined values across the four measures (note the missing values of $RangeEn_{A}$ as vacant points in the figures, especially in the white noise and pink noise results). Finally, the level of exponential decay in the frequency domain appears to be coded in the slope and starting point of the *RangeEn* trajectories in the *r*-plane. Figure 2 suggests that Brown noise ($1/f^{2}$, with the largest spectral decay amongst the three noise types) has the lowest entropy pattern, while white noise with no decay in the frequency domain has the steepest entropy trajectory, with the largest starting value of ≥4 at *r* = 0.

### 3.3. Dependency to Signal Amplitude

To evaluate the effect of signal amplitude on entropy, we simulated a white noise signal $x_{1}\left(n\right)$ with *N* = 1000 time points and its copy multiplied by 5, i.e., $x_{2}\left(n\right)=5x_{1}\left(n\right)$ (see first and second rows in the top panel of Figure 3, respectively). We then computed *ApEn*, *SampEn*, $RangeEn_{A}$, and $RangeEn_{B}$ for *m* = 2 and a range of tolerance values *r* from 0.01 to 1 with Δr = 0.01. As Figure 3 shows, $RangeEn_{A}$ and $RangeEn_{B}$ obtained from $x_{1}\left(n\right)$ and $x_{2}\left(n\right)$ are nearly identical, while *ApEn* and *SampEn* diverge. In most of the existing *ApEn* and *SampEn* studies in the literature, the input signal is divided by its SD to reduce the dependency of the entropy on the signal gain factor. This solution is useful only for stationary changes of signal amplitude, where the entire SD of the whole signal is an accurate description of its variability. We therefore designed a more difficult test for the entropies using a nonstationary signal $x_{3}\left(n\right)$, whose SD is time-varying:(23)$$x_{3}\left(n\right)=\left\{\begin{array}{ll} x_{1}\left(n\right) & n=1,\ldots ,200\\ 3x_{1}\left(n\right) & n=201,\ldots ,400\\ 10x_{1}\left(n\right) & n=401,\ldots ,600\\ 4x_{1}\left(n\right) & n=601,\ldots ,800\\ x_{1}\left(n\right) & n=801,\ldots ,1000. \end{array}\right..$$

The signal $x_{3}\left(n\right)$ (illustrated in the third row in the top panel of Figure 3) resembles a nonstationary random process which has been generated through a stationary process modelled by $x_{1}\left(n\right)$, but also affected by a time-varying amplitude change. In order to correct for the amplitude (gain) variation prior to computing the entropies *ApEn* and *SampEn*, we replaced $x_{3}\left(n\right)$ by $x_{3}\left(n\right)/\sigma_{x_{3}}$ for these two entropy measures where $\sigma_{x_{3}}$ is the standard deviation of $x_{3}\left(n\right)$. As entropy patterns of Figure 3 suggest, even after applying this amplitude correction, *ApEn* and *SampEn* are still sensitive to amplitude changes. This is, however, not the case for *RangeEn* measures that are much less affected by this nonstationary change.

### 3.4. Relationship with the Hurst Exponent

The results of *ApEn* and *SampEn* for *fLm* signals with different Hurst exponents are summarised in Figure 4. As seen in Figure 4A–D, *ApEn* and *SampEn* show a systematic relationship with the Hurst exponent. In particular, *SampEn* has an inverse monotonic relationship with the Hurst exponent in the *r*-plane (note the descending colour scale along the y-axis at all *r* values). Although the relationship between *ApEn* and Hurst is not as monotonic as that of *SampEn*, it still shows a systematic change. One way of quantifying these changes is by examining their corresponding *m*-exponents and *r*-exponents (i.e., the linear slopes of entropy patterns versus *ln(m)* and *ln(r)*, respectively. See Section 2.4.2 and Section 2.4.3). Figure 4E–H suggest that *m*- and *r*-exponents of both *ApEn* and *SampEn* are related to the Hurst exponent in a nonlinear way, before signal amplitude correction. Additionally, their trajectories reach a plateau in the *r* and *m* domains at high self-similarity levels (note the relatively flat regions of red dots in Figure 4E–H). This implies that *ApEn* and *SampEn* lose their link with the Hurst exponent in highly self-similar signals. The black dotted plots in Figure 4E–H suggest that signal amplitude correction results in a more linear relationship between the Hurst exponent and *r*-exponent, but it is less for the *m*-exponent.

We repeated the same analysis for *fBm* signals with the results illustrated in Figure 5. As the figure shows, signal amplitude correction has a more significant impact on entropy patterns of *fBm* than *fLm*. For example, there is almost no systematic relationship between *SampEn* and the Hurst exponent without amplitude correction (see Figure 5B versus Figure 5D). Additionally, the number of defined *SampEn* is reduced if we do not perform this correction (note the very low number of red dots in Figure 5F and the absence of any red dot in Figure 5H). Similar to the *fLm* results in Figure 4, amplitude correction can linearise the relationship between entropy exponents and the Hurst exponent.

A similar analysis using $RangeEn_{A}$ and $RangeEn_{B}$ highlights their properties in contrast to *ApEn* and *SampEn*. The results extracted from *fLm* and *fBm* signals are summarised in Figure 6 and Figure 7, respectively. Firstly, the patterns of $RangeEn_{A}$ and $RangeEn_{B}$ are relatively similar to each other, except that $RangeEn_{A}$ has a considerable amount of missing (undefined) values, specially over low *r* values. Secondly, the *r*- and *m*-exponents of both $RangeEn_{A}$ and $RangeEn_{B}$ have a more linear relationship with the Hurst exponent than *ApEn* and *SampEn*. In particular, the flat regions of their exponents over high *H* values are shorter than those of *ApEn* and *SampEn* (see Panels E to H of Figure 6 and Figure 7).

### 3.5. Linear Scaling of the Covariance Matrix in fBm

As another test of robustness to amplitude variations, we investigated whether the relationship between signal entropy and Hurst exponents of *fBm* (Figure 5 and Figure 7) are independent from linear scaling of its covariance matrix defined in Equation (Equation 12). We simulated *fBm* signals using the Cholesky decomposition method [37] (fbm function of Python’s *nolds* library) at a Hurst exponent of *H* = 0.75 and five scaling coefficients *D* = 0.001, 0.01, 1, 10, and 100, where the value of *D* = 1 leads to the original form of *fBm*. Figure 8 shows the estimated entropy patterns of altered *fBm*. Note that we did not correct the input signals to *ApEn* and *SampEn* by their standard deviation to ensure the same testing condition for all four measures. The results in Figure 8 show that *ApEn* and *SampEn* are more vulnerable to linear scaling of the covariance matrix than *RangeEn*.

### 3.6. Analysis of Epileptic EEG

We performed self-similarity analysis of epileptic EEG datasets by extracting their Hurst exponent through the standard rescaled range approach [38] (hurst_rs function of Python’s *nolds* library). Figure 9A illustrates the distributions of Hurst exponents for three datasets. Whilst interictal segments are clustered toward higher self-similarity levels, ictal segments have been distributed across a wider range between high and low self-similarity. Figure 9B–E represent the patterns of *ApEn*, *SampEn*, $RangeEn_{A}$, and $RangeEn_{B}$ in the *r*-plane for the three EEG datasets (corrected amplitudes and fixed *m* of 2). In all plots, the two interictal *r*-trajectories are close to each other and represent a relatively different trajectory to the ictal state.

## 4. Discussion

In this study, we showed that signal complexity measures of *ApEn* and *SampEn* are linked to the self-similar properties of signals quantified by their Hurst exponent. However, they may become insensitive to high levels of self-similarity due to the nonlinear nature of this relationship. We subsequently introduced a modification to *ApEn* and *SampEn* (called RangeEn) that not only improves their insensitivity issue but also alleviates the need for amplitude correction.

Signal complexity analysis can be approached through the concept of state vectors in the reconstructed phase space [23,30,39]. From this perspective, *ApEn* and *SampEn* of a random process assess its dynamics in the phase space by quantifying the evolution of its states over time. This is done through computing the Chebyshev distance $d_{chebyshev}$, as a measure of similarity between state vectors, and obtaining the conditional probability of space occupancy by the phase trajectories, as detailed in Section 2.1.1, Section 2.1.2 and Section 2.1.3. However, $d_{chebyshev}$ only considers the maximum element-wise difference between two state vectors while ignoring the lower limit of this differences. In addition, it is not normalised, thus sensitive to changes in signal magnitude (gain) and defined for all values of the tolerance parameter *r* (from 0 to *∞*). This last issue leads to unbounded values of *ApEn* and *SampEn* along the *r*-axis. In order to alleviate these limitations, we replaced $d_{chebyshev}$ with a normalised distance (called range distance or $d_{range}$) defined in Equation (Equation 17) prior to computing the entropies *ApEn* and *SampEn*. This led to modified forms of *ApEn* and *SampEn*, namely $RangeEn_{A}$ and $RangeEn_{B}$, respectively.

$RangeEn_{A}$ and $RangeEn_{B}$ offer a set of desirable characteristics when applied to simulated and experimental data. First, they are more robust to signal amplitude changes compared to *ApEn* and *SampEn*. This property originates from the fact that the distance used in the definition of the proposed entropies is normalised between 0 and 1. Unlike *ApEn* and *SampEn* measures that require an extra amplitude regulation step that involves multiplying the tolerance parameter *r* by the input signal’s standard deviation [24], the *RangeEn* measures are needless of any amplitude correction. This is a plausible feature when analysing real-world signals, which are usually affected by confounding amplitude changes such as artefacts. Figure 3 illustrates two situations where *ApEn* and *SampEn* are highly sensitive to variations of signal amplitude, contrary to *RangeEn* measures. It is for future work to investigate the vulnerability of RangeEn to more complicated cases of nonstationarity compared with those shown in Figure 3.

The second desirable property of *RangeEn* is that, regardless of the dynamic nature of the signal, both $RangeEn_{A}$ and $RangeEn_{B}$ measures always reach 0 at the tolerance value *r* of 1. The explanation of this property is straightforward: *r* = 1 is the value where all *m*-long segments $X_{i}^{m}$ and $X_{j}^{m}$ match. This leads to the joint conditional probability being 1 (see Equations (Equation 19)–(Equation 22)).

According to the simulation results of *fLm* and *fBm* signals with certain Hurst exponents, all of *ApEn*, *SampEn*, and *RangeEn* measures are able to reflect the self-similarity of time series to different extents. However, *RangeEn* have a more linear relationship with the Hurst exponent. This brings us to the third property of the *RangeEn* measures, namely a more linear link between their *r*- and *m*-exponents and the Hurst exponent, compared to *ApEn* and *SampEn*. We evaluated this property by extracting *RangeEn* measures from *fLm* and *fBm* signals, as their level of self-similarity can be accurately controlled through their Hurst exponent. We simulated these processes for different values of the Hurst exponents ranging from 0.01 (very short memory or high anti self-similarity) to 0.99 (very long memory or high self-similarity). The simulation results (Figure 4, Figure 5, Figure 6 and Figure 7) reveal a regular pattern of almost linearly decreasing Hurst exponents associated with the slope of *RangeEn* trajectories versus *ln(r)* and *ln(m)*. This pattern is more nonlinear and sometimes non-monotonically increasing for *ApEn* and *SampEn*.

Among the four signal entropy measures investigated in our study, *ApEn* is the only measure that is always defined due to the self-matching of state vectors (or templates) in its definition [23]. *SampEn* and $RangeEn_{B}$ may result in undefined values, as they compute the logarithm of the sum of conditional probabilities $C_{i}^{m}\left(r\right)$, which could lead to *ln(0)* (see Section 2.1.3 and Section 2.3.1 for more details). This issue may also happen to $RangeEn_{A}$, as it calculates the sum of log probabilities (i.e., $ln\;C_{i}^{m}\left(r\right)$). However, the number of undefined values in $RangeEn_{A}$ is usually much higher than *SampEn* and $RangeEn_{B}$. This is because it is more likely that all joint conditional probabilities ($C_{i}^{m}\left(r\right)$) between a single state vector and the rest of the state vectors in the phase space become zero, in particular, at small tolerance values of *r* where the small partitions of phase space are not visited by any trajectory. Figure 7 provides an exemplary situation where there are many undefined $RangeEn_{A}$ values for *fBm* compared to the other three.

A realisation of real-world signal complexity is reflected in EEG signals. EEG conveys information about electrical activity of neuronal populations within cortical and sub-cortical structures in the brain. Epilepsy research is a field that significantly benefits from EEG analysis, as the disease is associated with abnormal patterns in EEG such as seizures and interictal epileptiform discharges [40]. Therefore, characterisation of abnormal events in epileptic EEG recordings is helpful in the diagnosis, prognosis, and management of epilepsy [27,41,42]. Our results suggest that interictal EEG at the intracranial level is more self-similar than ictal EEG with clustered Hurst exponents toward 1. On the other hand, the Hurst exponent of ictal EEG covers high and low self-similarity (see Figure 9A). Therefore, self-similarity may not be a discriminative feature of ictal state. All entropy measures represent distinctive trajectories in the *r*-plane for interictal versus ictal states with relatively low variance over EEG segments (see Figure 9B–E). This implies that signal complexity analysis may be more beneficial than self-similarity analysis for epileptic seizure detection and classification. Note that, in the absence of any time-varying artefact, EEG signals can be considered as weak stationary processes [43]. Therefore, a correction of the amplitude changes by the standard deviation of the signal in *ApEn* and *SampEn* may lead to comparable results with *RangeEn*.

Multiscale entropy is a generalisation of *SampEn* where the delay time (or scale factor) τ in Equation (Equation 1) is expanded to an interval of successive integers starting from 1 through coarse-graining of the input signal [25]. It is straightforward to extend this idea to the *RangeEn* measures. Exploring the properties and capacities of multiscale *RangeEn* is left for future research.

## 5. Conclusions

In this study, we proposed modifications to *ApEn* and *SampEn* called $RangeEn_{A}$ and $RangeEn_{B}$, respectively. We showed that these new signal complexity measures, compared with *ApEn* and *SampEn*, are more sensitive to self-similarity in the data and more robust to changes in signal amplitude. Additionally, they do not need any signal amplitude correction. We showed, in an exemplary application, that signal entropies can differentiate between normal and epileptic brain states using EEG signals. Given the high interest accorded to *ApEn* and *SampEn* in different scientific areas (more than 4000 citations each since being introduced), we believe that our present study has targeted a significant problem by addressing some of their important practical limitations.

## Figures and Tables

**Figure 1 entropy-20-00962-f001:**
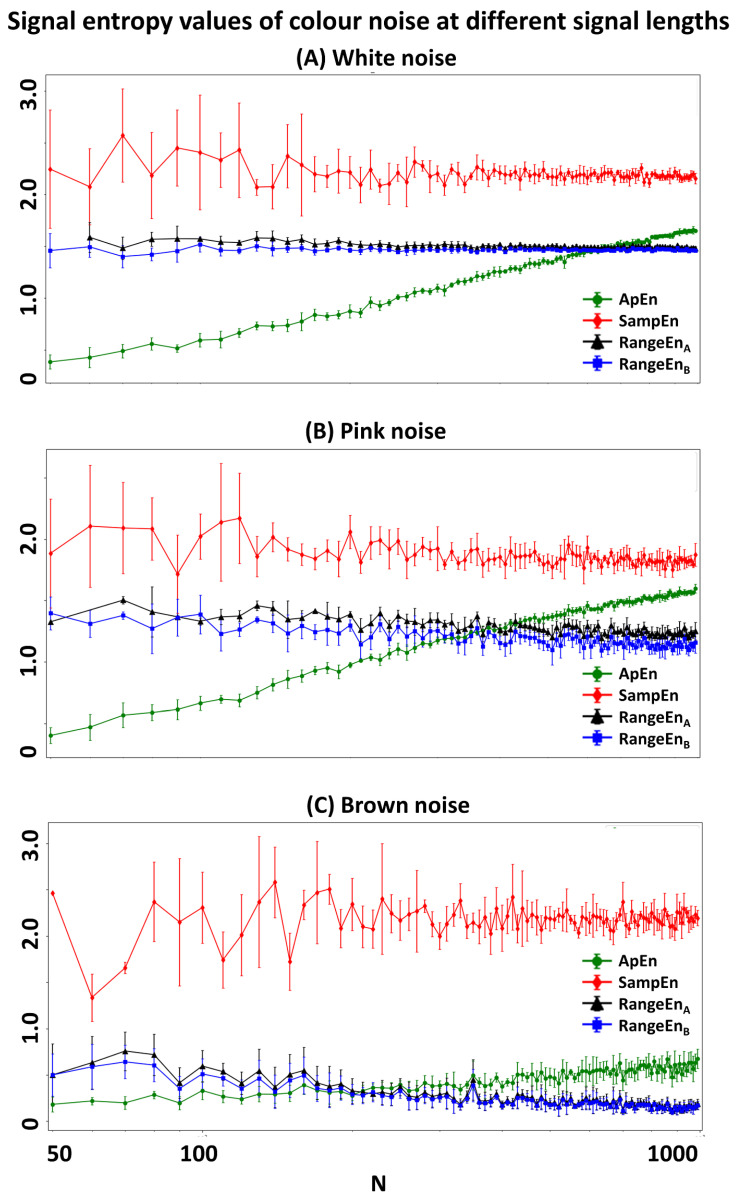
Variation of the entropy measures over different signal lengths (*N* in time samples). Each noise type has been simulated 100 times, and errorbars represent the variation over noise realisations. $RangeEn_{A}$ (in black) and $RangeEn_{B}$ (in blue) show less deviation around their mean values compared to *ApEn* (in green) and *SampEn* (in red), in particular over short signal lengths. In all panels, the x-axis is on a logarithmic scale.

**Figure 2 entropy-20-00962-f002:**
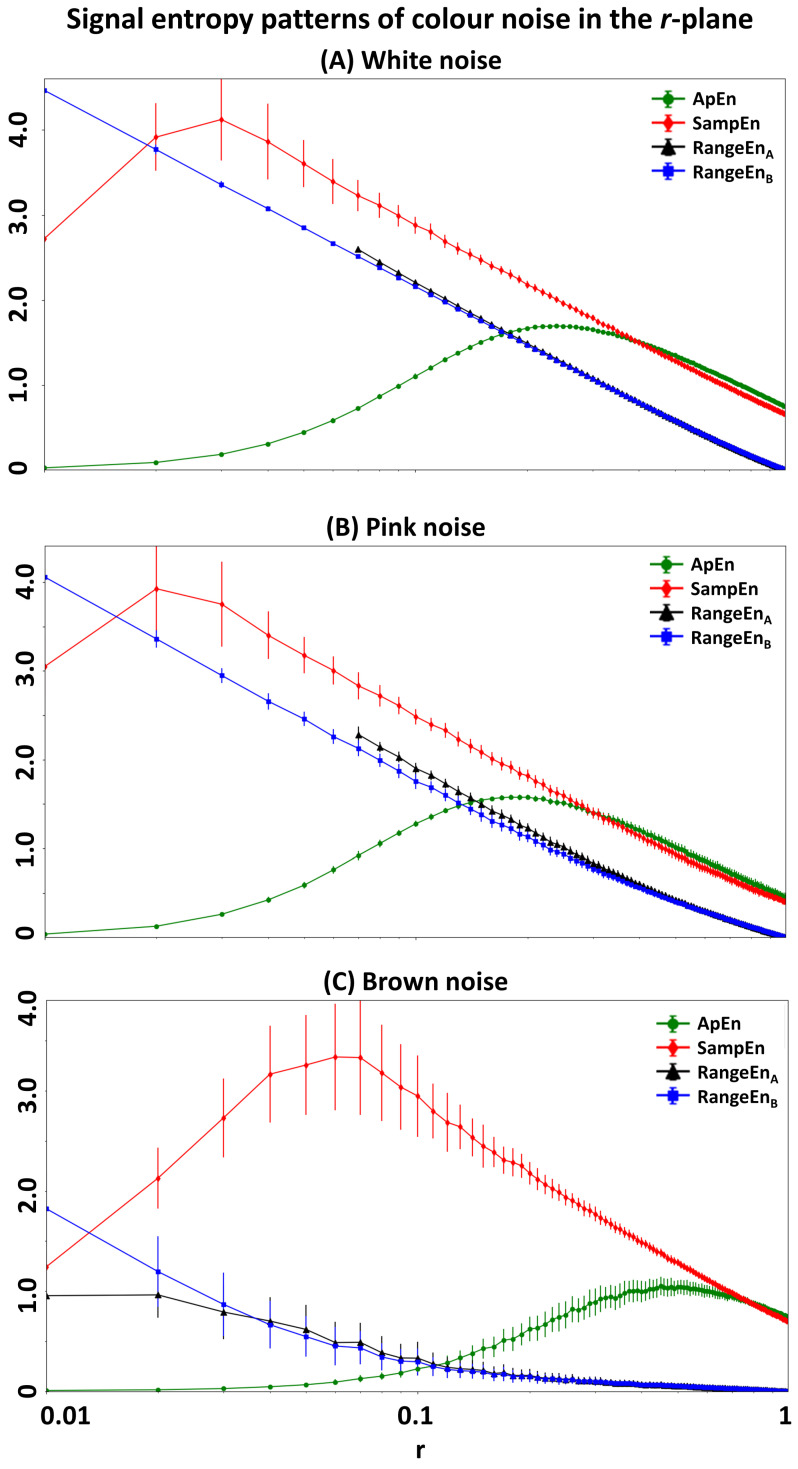
Impact of the tolerance parameter *r* on the signal entropy measures extracted from three noise types. Note that *RangeEn* measures always reach to 0 at *r* = 1, but this is not the case for *ApEn* and *SampEn*. In all panels, entropy measures have been illustrated in distinct colors and the x-axis is on a logarithmic scale.

**Figure 3 entropy-20-00962-f003:**
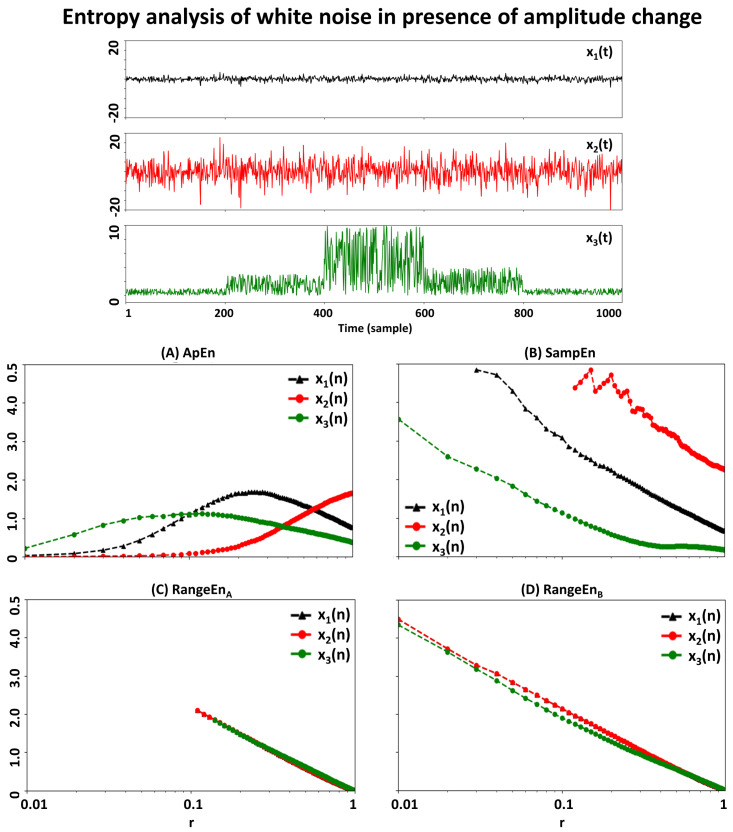
Dependency of the signal entropy measures to stationary and nonstationary amplitude changes. Top panel shows three input signals, i.e., white noise ($x_{1}\left(t\right)$, in black), scaled white noise by a constant coefficient ($x_{2}\left(t\right)=5x_{1}\left(t\right)$, in red), and scaled white noise by a time-varying coefficient ($x_{3}\left(t\right)$ defined in Equation (Equation 23), in green). Panels A–D demonstrate the signal entropy trajectories over the *r* interval of 0.01 to 1, with 0.01 increasing steps. Note that the patterns of $RangeEn_{A}$ and $RangeEn_{B}$ are almost identical for white noise and both of its scaled versions, but *ApEn* and *SampEn* show drastic changes after any change in the amplitude of their input signal.

**Figure 4 entropy-20-00962-f004:**
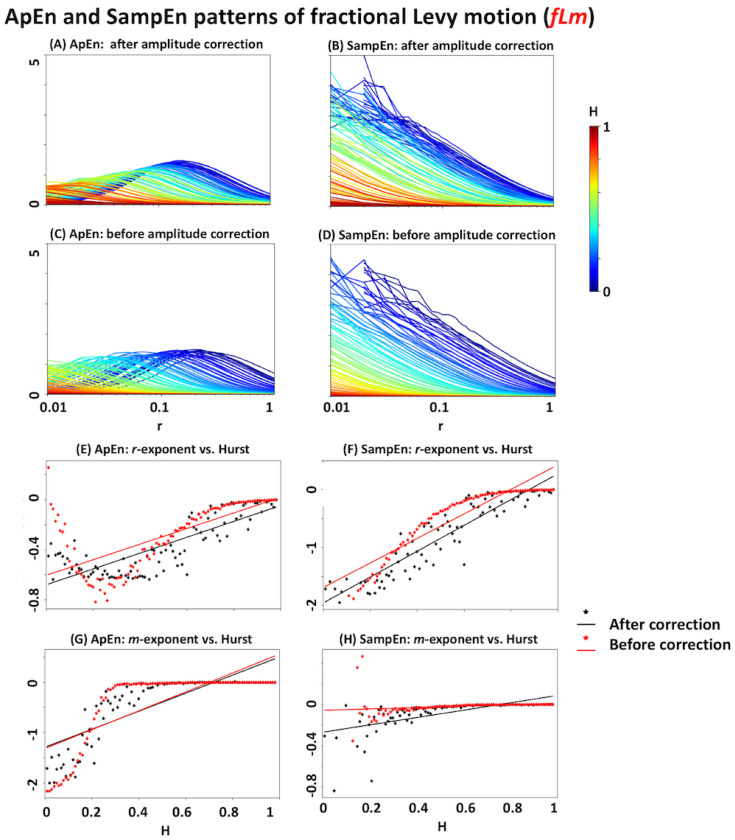
*ApEn* and *SampEn* analyses of fractional Levy motion (*fLm*). Panels (**A**–**D**) illustrate the entropy trajectories in the *r*-plane with pre-defined Hurst exponents ranging from 0.01 to 0.99 with increasing steps of ΔH = 0.01. Each analysis has been repeated for two conditions: with and without amplitude correction (i.e., dividing the input signal by its standard deviation). The *H* values have been colour-coded. The missing points in each plot have been left as blank. In all panels, the x-axis is on a logarithmic scale. Panels (**E**) and (**F**) represent the scatter plots of *r*-exponents (i.e., the slope of the fitted line to the measure versus *ln(r)*), before and after amplitude correction. Panels (**G**) and (**H**) represent the scatter plots of *m*-exponents (i.e., the slope of the fitted line to the measure versus *ln(m)*), before and after amplitude correction.

**Figure 5 entropy-20-00962-f005:**
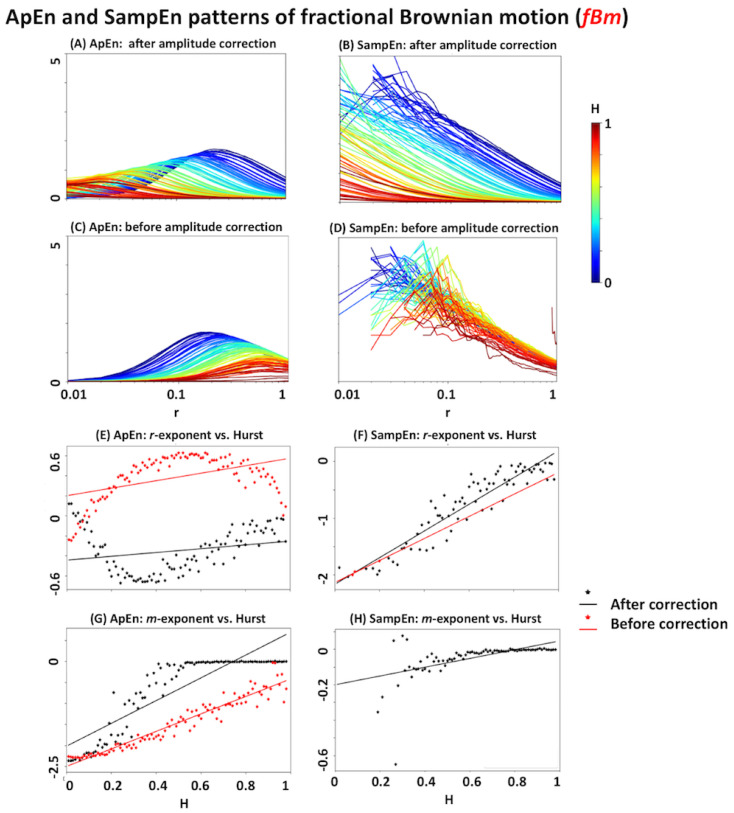
*ApEn* and *SampEn* analyses of fractional Brownian motion (*fBm*). See the caption of Figure 4 for more details.

**Figure 6 entropy-20-00962-f006:**
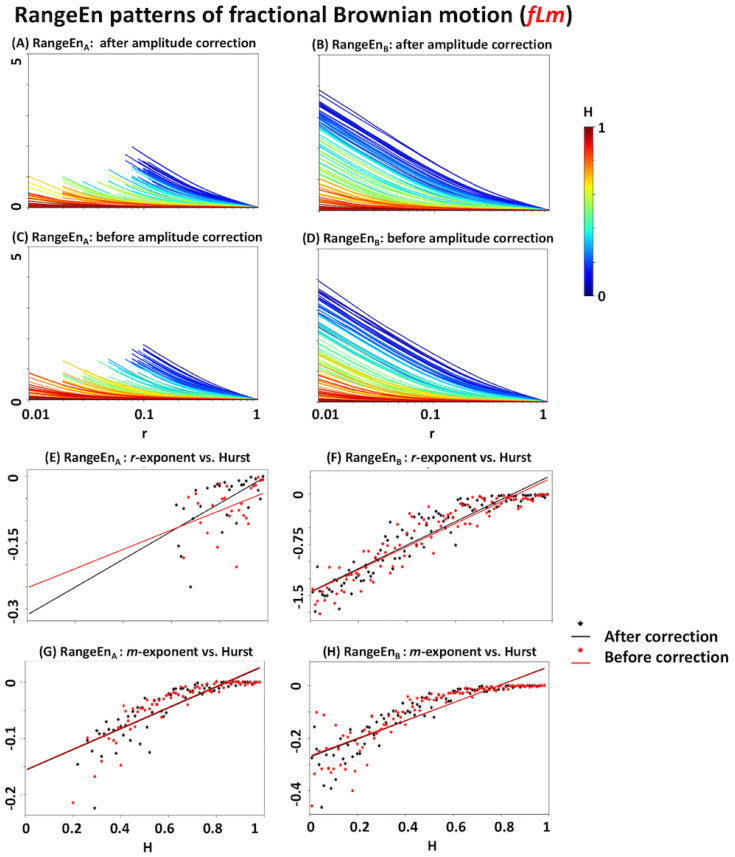
Analyses of $RangeEn_{A}$ and $RangeEn_{B}$ of fractional Levy motion (*fLm*). See the caption of Figure 4 for more details.

**Figure 7 entropy-20-00962-f007:**
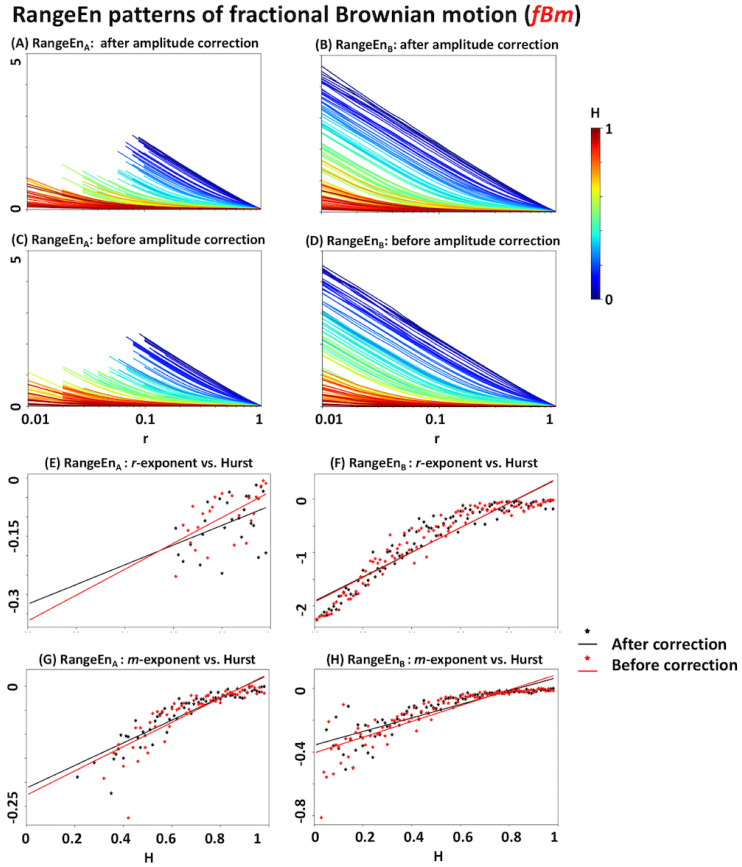
$RangeEn_{A}$ and $RangeEn_{B}$ analyses of fractional Brownian motion (*fBm*). See the caption of Figure 4 for more details.

**Figure 8 entropy-20-00962-f008:**
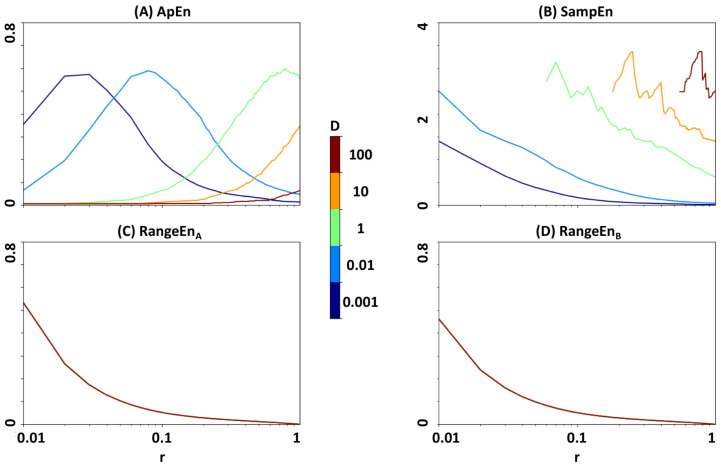
Sensitivity of the entropy measures to linear scaling of the covariance matrix in *fBm* (see also Equation (Equation 12)). Panels (**A**–**D**) illustrate the entropy trajectories in the *r*-plane at *H* = 0.75 and five scaling factors of *D* = 0.001, 0.01, 1, 10, and 100. The *D* values have been colour-coded. The missing points in each plot have been left as blank.

**Figure 9 entropy-20-00962-f009:**
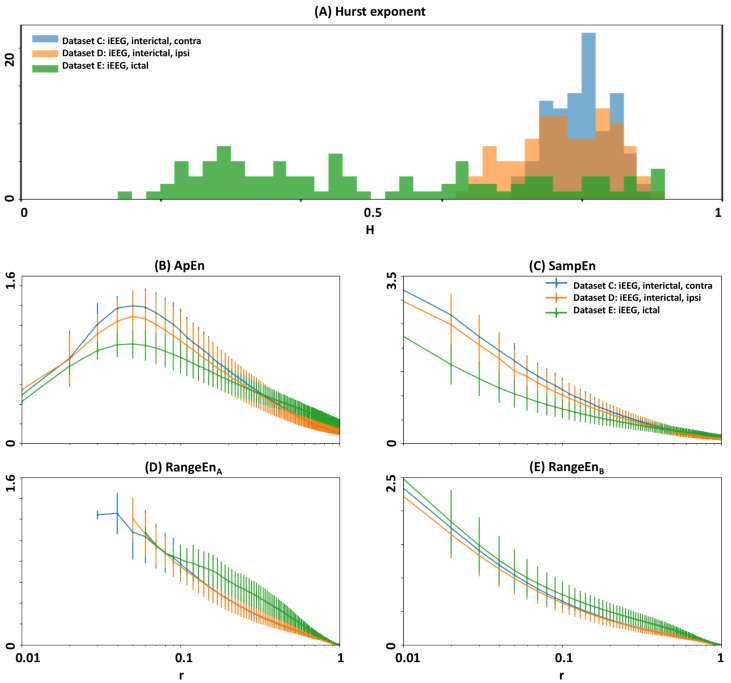
Self-similarity and complexity analyses of epileptic EEG. Datasets C, D, and E have been taken from the public EEG database of [36]. In the legends, iEEG stands for intracranial EEG. (**A**) Distributions of the Hurst exponent extracted from EEG segments. (**B**–**E**) Trajectories of *ApEn*, *SampEn*, $RangeEN_{A}$, and $RangeEn_{B}$, respectively. For all entropy measures, the embedding dimension parameter *m* was fixed to 2. In each plot, error bars show one standard deviation.

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
