# Peer review of "Range Entropy: A Bridge between Signal Complexity and Self-Similarity"

_entropy, 2018, doi:10.3390/e20120962_

Reviewer 1 Report

SampEn and ApEn are information measures commonly used in the analysis of different biological signals to establish the complexity of the corresponding time series. However, it still suffers from issues such as inconsistency over short-size signals and its tolerance parameter r, susceptibility to signal amplitude changes and insensitivity to self-similarity of time series. The authors set their own task: (i) to modify the ApEn and SampEn measures, which are defined for 0 < r < 1, and (ii) that new signal entropy measures (which are defined for 0< r << span="">1) has to be more robust to signal amplitude changes and sensitive to self-similarity property of time series. They do that through the following steps: (i) modification of ApEn and SampEn by redefining the distance function used originally in their definitions, (ii) evaluation of the new entropy measures, called range entropies (RangeEn) using different random processes and nonlinear deterministic signals. The aim of this paper was to study the effects of signal length, amplitude changes, and self-similarity on the new entropy measures. Finally, the authors applied the proposed entropies to normal and epileptic electroencephalographic (EEG) signals under different states.

In assessing this manuscript, I had a small dilemma about how to do it. However, I decided to do the assessment in the following way. It is obvious that the authors have had measured biological time series that indicated that something more radical has to be corrected in the often used informational measures (SampEn and ApEn) in the estimation of order/disorder in those series. That's how entropy arose the RangeEn. Instead of a more rigorous mathematical elaboration, the authors decided to justify their information measure through several examples of deterministic maps. This is also a regular way to convince the reading community. I have nothing against that but authors in their mathematical elaboration produced text like the text in the textbook. Therefore, some parts of the manuscript have the form of the textbook. Finally, I stand at the point of view that after revision (major or minor; finally it does not matter) this manuscript can be recommended for the publication.

Remarks

# the title covers only part of what has been done in the manuscript. In this form the title can lead the reader to the direction which is not strongly elaborated in this version of the paper;

# abstract too long (more than 200 words) – 325 words. Please, bring it to the number required by the Entropy rules;

# Between lines 55-76 is placed the text, which unnecessarily burdens the text.  In addition, it is not directly used in the further text.

# Line 188: What about the transition domain in the logistic equation.

Author Response

In assessing this manuscript, I had a small dilemma about how to do it. However, I decided to do the assessment in the following way. It is obvious that the authors have had measured biological time series that indicated that something more radical has to be corrected in the often used informational measures (SampEn and ApEn) in the estimation of order/disorder in those series. That's how entropy arose the RangeEn. Instead of a more rigorous mathematical elaboration, the authors decided to justify their information measure through several examples of deterministic maps. This is also a regular way to convince the reading community. I have nothing against that but authors in their mathematical elaboration produced text like the text in the textbook. Therefore, some parts of the manuscript have the form of the textbook. Finally, I stand at the point of view that after revision (major or minor; finally it does not matter) this manuscript can be recommended for the publication.

Response: Thank you for the supportive feedback and helpful remarks. In order to reduce the similarity of our manuscript with textbooks, we decided to omit sections outlining the basic and unnecessary explanations of Shannon entropy, Kolmogorov-Sinai entropy and Eckmann-Ruelle entropy. Instead we point the reader towards key papers in the literature. We have also modified the general structure of the manuscript by adding a Materials and Methods section to that and rearranging several sub-sections in Results. This is also in line with your third comment.

Comment 1: The title covers only part of what has been done in the manuscript. In this form the title can lead the reader to the direction which is not strongly elaborated in this version of the paper;

 Response and action: We agree that the title ‘Range entropy: A bridge between signal complexity and self-similarity’ was not precise enough for the previous version of the manuscript. In the updated version, however, we have added more rigorous and comprehensive analyses about the relationship between signal complexity and self-similarity using two types of stochastic processes (in contrast to only one type in the initial version). Therefore, we feel that the title is now more appropriate and informative for the readership.

Comment 2: Abstract too long (more than 200 words) – 325 words. Please, bring it to the number required by the Entropy rules;

Response and action: Thank you for raising this important issue. We have now cut the Abstract to less than 200 words (page 1, lines 1-13):

Approximate entropy (ApEn) and sample entropy (SampEn) are widely used for temporal complexity analysis of real-world phenomena. However, their relationship with the Hurst exponent as a measure of self-similarity has not been deeply studied. Also, ApEn and SampEn suffer from susceptibility to signal amplitude changes. A common practice for addressing this issue is to correct their input signal amplitude by its standard deviation. In this study, we first show using simulations that ApEn and SampEn are related to the Hurst exponent through their tolerance and embedding dimension parameters. We then propose a modification to ApEn and SampEn called Range entropy or RangeEn. We show that RangeEn is more robust to nonstationary changes in signals and is more linearly related to the Hurst exponent, compared to ApEn and SampEn. RangeEn is bounded in the tolerance r-plane between 0 (maximum entropy) and 1 (minimum entropy) and has no need for signal amplitude correction. Finally, we demonstrate the usefulness of signal entropy measures for characterization of epileptic EEG data as a real-world example. A Python implementation of RangeEn is publicly available at https://github.com/omidvarnia/RangeEn.

Comment 3: Between lines 55-76 is placed the text, which unnecessarily burdens the text.  In addition, it is not directly used in the further text.

Response and action: Thank you for the useful suggestion. In the revised version, we have excluded the sections related to Shannon, Kolmogorov-Sinai and Eckmann-Ruelle entropies and instead, have added a new section called ‘Reconstructed phase space’.

Comment 4: Line 188: What about the transition domain in the logistic equation.

Response and action: Again, we chose to remove the deterministic and chaotic models from the updated study, because we did not really have a rigorous justification and suitable interpretation of the results. Investigation of these processes will be left for our future work.

Reviewer 2 Report

In this paper the Authors are proposing modifications to the approximate entropy (ApEn) and Sample entropy (SampEn) measures.

They have shown the effects of signal length, amplitude changes, and self-similarity on the new entropy measures.

In the paper they have applied the ApEn, the SampEn and the range entropies (RangeEn) to normal and epileptic electroencephalographic (EEG) signals under different states. They have calle the new new signal entropy measures RangeEnA and RangeEnB. The results indicate that a better performance on the EEG analysis results can be reached using the RangeEn.

After carefully reading, I find that this paper is extremely interesting, however in order to further improve I would only recommend to improve the references on the background. (I suggest: doi: 10.3390/e17020560, doi: doi.org/10.3390/e19090475, doi: 10.3390/e16126573 doi: 10.3390/e19070291 doi: 10.3390/e17010231,)

Author Response

Response and action: We wish to thank the reviewer for their kind comments on our manuscript. The five references you suggested above have been added to the first paragraph of the Introduction section (references 1, 2, 3, 10, 11). It now reads (page 1, lines 17-19):

Complexity is a global concept in data analysis that is observed in a wide range of real-world phenomena and systems including biological signals [16], brain dynamics [79], mechanical systems [10,11], climate change [12], volcanic eruption [13], earthquakes [14] and financial markets [15].

Reviewer 3 Report

The article discusses a new type of entropy for signal analysis. The authors postulate that the newly proposed entropy measure eliminates a range of shortcomings in other measures (e.g. approximate and sample entropy). The article is readable, well structured. I consider the use of simulation data together with EEG signals to be sufficient. The analysis of the comparison of different entropy measurements on data supports the author's claim about the benefits of the newly defined measure. I am not an EEG expert, but arguments about the use of distance entropy for analysis in epilepsy research seems to be well-founded and sufficient.

Minor remarks:

- There are some misspellings in the text, for example, line 34 Kulmogorov

- A brief summary (conclusion) section would be beneficial

- The introduction could be more elaborate

Author Response

The article discusses a new type of entropy for signal analysis. The authors postulate that the newly proposed entropy measure eliminates a range of shortcomings in other measures (e.g. approximate and sample entropy). The article is readable, well structured. I consider the use of simulation data together with EEG signals to be sufficient. The analysis of the comparison of different entropy measurements on data supports the author's claim about the benefits of the newly defined measure. I am not an EEG expert, but arguments about the use of distance entropy for analysis in epilepsy research seems to be well-founded and sufficient.

 Response: We appreciate the positive feedback and useful comments of the reviewer.

Comment 1: There are some misspellings in the text, for example, line 34 Kulmogorov

 Response and action: Thank you for drawing our attention to this misspelling. We have gone through the whole manuscript and corrected all typos.

Comment 2: A brief summary (conclusion) section would be beneficial

 Response and action: we agree with this comment. We have now added the following Conclusion section at the end of the manuscript (page 12, lines 381-390):

 In this study, we addressed shortcomings of ApEn and SampEn by proposing two alternatives named RangeEnA and RangeEnB. We showed that these new signal complexity measures, compared with ApEn and SampEn, are more sensitive to self-similarity in the data and more robust to changes in signal amplitude. Also, they do not need any signal amplitude correction. We showed, in an exemplary application, that signal entropies can differentiate between normal and epileptic brain states using electroencephalographic (EEG) signals. Given the high interest accorded to ApEn and SampEn in different scientific areas (more than 4000 citations each since being introduced), we believe that our present study has targeted a significant problem by addressing some of their important practical limitations.

Comment 3: The introduction could be more elaborate

 Response and action: We have added a new opening paragraph to the Introduction section (page 1, lines 17-32):

 Complexity is a global concept in data analysis that is observed in a wide range of real-world phenomena and systems including biological signals [16], brain dynamics [79], mechanical systems [10,11], climate change [12], volcanic eruption [13], earthquakes [14] and financial markets [15]. It is difficult to provide a formal definition for signal complexity. This concept, however, can be approached as a mid-point situation between signal regularity and randomness. From this perspective, complexity can be defined as the amount of information that a time series conveys over time. Highly random fluctuations (such as white noise as an extreme case) have very low complexity, because they present no regular pattern in their dynamical behavior. Real-world phenomena, on the other hand, usually contain spreading patterns of nonlinear ’structured activity’ across their frequency components and temporal scales. The dynamic behavior of the brain or fluctuation of stock markets are examples of complex processes. Despite the importance of complexity in science, its quantification is not straightforward. Time-frequency distributions and wavelet transforms [16] are examples of analysis tools for capturing signal dynamics, but they may be insensitive to nonlinear changes.

A complementary approach to time-frequency distributions and wavelet transforms is signal entropy analysis, a family of methods rooted in information theory.

We have also updated the last two paragraphs in this section as follows (page 2, lines 43-59):

 Among these signal entropy measures, ApEn and SampEn are two of the most commonly used measures in contemporary science, especially in the analysis of biological signals [27]. Like ApEn, SampEn resembles a template-matching search throughout the input signal with two main parameters: embedding dimension m and tolerance r. The former governs the length of each segment (template) to be searched and the later controls the level of similarity between segments. In fact, SampeEn stems from ApEn after addressing some of its limitations including inconsistency over the parameter r and strong dependency to the input signal length [24]. However, both measures still suffer from sensitivity to signal amplitude changes. Another important aspect of these measures is their inverse relationship with the Hurst exponent as a measure of self-similarity in signals [28]. The analysis of this link, however, deserves more attention.

In this study, we investigate the behaviour of ApEn and SampEn in presence of self-similarity and examine their relationship with the Hurst exponent through their tolerance and embedding dimension parameters. We also address the issue of sensitivity to signal amplitude changes in ApEn and SampEn and develop modified versions called range entropies or RangeEn. We compare RangeEn with ApEn and SampEn from different perspectives using multiple simulations. Finally, we demonstrate the capacity of signal entropy measures for epileptic EEG characterization.

Reviewer 4 Report

This paper offers two new definitions for the information generation rate of a time series based on a slight change to the definitions for the approximate entropy and sample entropy.  The authors argue that these new definitions are more sensitive to self-similarity in the data, amongst other benefits, including robustness to changes in signal amplitude and lower amounts of data required to correctly estimate the entropies.

I found the paper to be interesting.  I'm not sure I am convinced of the utility of these new definitions yet, but I think it's a direction worth pursuing.  To that end, I have a few comments that I hope will enable the authors to better convince readers like me that they should try using these definitions to characterize our time series.

Major comment:

I am very unconvinced by the simulation-based proof that the new "distance" leads to a connection between the new entropies and the Hurst exponent.  First, a minor point: one can calculate the Hurst exponent of a renewal process and find some answer between 0 and 1, depending on the waiting time distribution; but if I understand the new distance definition correctly, the RangeEns will be nearly constant in r (due to the discreteness of the possible entries of x) and will not scale with the Hurst exponent.  So it may be that the relationship between Hurst exponent and these new entropies only holds when the process is continuous-valued.  But even then, I'm unconvinced.  The fBm is parametrized only by the Hurst exponent, so it is of little surprise that ApEn and both RangeEns all showed a clear one-to-one relationship between their f_m(r) and H.  Suppose you slightly alter the fBm so that the covariance function is multiplied by D, where D is some parameter, i.e.

E[B(t) B(s)] = D*(1/2)*(t^{2H}+s^{2H}-|t-s|^{2H}),

and redo the analysis.  I have a feeling that the ApEn will be sensitive to D, but both RangeEns will not be sensitive to D due to the in-built normalization, and so a slightly more convincing case will be made that RangeEns alone correlate with Hurst exponents.  But let's go beyond the in-built normalization invariance.  I would recommend trying to relate the Hurst exponent and RangeEns for a fractional Levy process (as one can vary the parameters of the integrating Levy process while retaining the Hurst exponent).  If you can show a one-to-one relationship between f_m(r) and H for those processes as you sweep over all possible parameters, I'll believe your claim.

Mid-level comments:

-- Eq. 16: I think it'd be good to point out ways in which this is intuitively not a distance.  (You point out ways in which it is formally not a distance.)  The d_range need not increase as X and X' get farther away from one another.  For instance, take some X and increase all of its component's distances from X' by some epsilon.  Then the numerator of d_range is equivalent, while the denominator increases, and so d_range decreases, even though most would say that X and X' are now farther apart.

-- There are many entropy-based measures that you did not mention that one could use to characterize a time series, though these measures are not as well-studied as those you did mention.  See "Anatomy of a Bit" by R. G. James et al.

-- For the Entropy analysis of EEG data, it seems to me that it might be easy to find something that classifies the time series into healthy, interictal, or ictal.  To that end, I would ask that you calculate the Hurst exponent of the EEG data and see if it clusters; and more than that, also calculate ApEn and sample entropy of the EEG data to see if it clusters by those measures too.

Minor comments:

-- Line 27: I'd say the entropy rate, rather than the entropy.  It would be a shame if the first sentence were to confuse the reader between the single-symbol entropy and the entropy rate.

-- Line 30: It's not a uniform probability density-- it's a normal distribution.

-- Line 34: Kolmogorov, not Kulmogorov.  Also, at the end of this paragraph, it might be worth mentioning that in Section 2, you will explain how all of these relate.

-- Line 52: Not a fan of the phrase "have better consistency", since I don't think it's clear what you mean.  Do you mean that N need not be as large in order to be close to the N-->infinity answer?

-- Eq. 2 gives the single-symbol entropy, whereas I think you want the formula for the entropy rate, which requires conditioning on pasts of symbols.

-- In Line 67, what do you mean by "can be quantified by K in Eq. 2"?  The K is merely a constant to go between different logarithmic bases.

-- Line 68: Might want to mention that Pesin's theorem is useful only for deterministic dynamical systems.  As a result, I also think that here you should discuss the epsilon-tau entropy rate and give the definition of the K-S entropy (rate) in terms of the epsilon, tau entropy rate, and then cite Pesin's theorem and give some of its conditions so that its limitations are clear.

-- Line 75: Again, do you mean single-symbol entropy or entropy rate?

-- Line after 76: "spacial" should be special

-- First paragraph of Section 3: "behviour" should be "behavior" and "centred" should be "centered"

Author Response

This paper offers two new definitions for the information generation rate of a time series based on a slight change to the definitions for the approximate entropy and sample entropy.  The authors argue that these new definitions are more sensitive to self-similarity in the data, amongst other benefits, including robustness to changes in signal amplitude and lower amounts of data required to correctly estimate the entropies.

I found the paper to be interesting.  I'm not sure I am convinced of the utility of these new definitions yet, but I think it's a direction worth pursuing.  To that end, I have a few comments that I hope will enable the authors to better convince readers like me that they should try using these definitions to characterize our time series.

 Response: We appreciate your constructive feedback and insightful comments. We agree with your concerns and have further clarified our proposal in the updated manuscript. 

Comment 1: I am very unconvinced by the simulation-based proof that the new "distance" leads to a connection between the new entropies and the Hurst exponent.  First, a minor point: one can calculate the Hurst exponent of a renewal process and find some answer between 0 and 1, depending on the waiting time distribution; but if I understand the new distance definition correctly, the RangeEns will be nearly constant in r (due to the discreteness of the possible entries of x) and will not scale with the Hurst exponent.  So it may be that the relationship between Hurst exponent and these new entropies only holds when the process is continuous-valued.  But even then, I'm unconvinced.  The fBm is parametrized only by the Hurst exponent, so it is of little surprise that ApEn and both RangeEns all showed a clear one-to-one relationship between their f_m(r) and H.  Suppose you slightly alter the fBm so that the covariance function is multiplied by D, where D is some parameter, i.e.

E[B(t) B(s)] = D*(1/2)*(t^{2H}+s^{2H}-|t-s|^{2H}),

and redo the analysis.  I have a feeling that the ApEn will be sensitive to D, but both RangeEns will not be sensitive to D due to the in-built normalization, and so a slightly more convincing case will be made that RangeEns alone correlate with Hurst exponents. 

 Response and action: We tested the idea of multiplying the fBm’s covariance matrix with a constant coefficient D. The results confirmed your hypothesis as ApEn and SampEn are sensitive to this change, but RangeEn’s are not. The new section 3.5 has been dedicated to his analysis which reads as follows (page 10, lines 281-289):

3.5. Linear scaling of the covariance matrix in fBm

As another test of robustness to amplitude variations, we investigated whether the relationship between signal entropy and Hurst exponents of fBm (FIGs. 5 and 7) are independent from linear scaling of its covariance matrix defined in Eq. 12. We simulated fBm signals using the Cholesky decomposition method [38] at a Hurst exponent of H=0.75 and five scaling coefficients D=0.001, 0.01, 1, 10, 100 where the value of D=1 leads to the original form of fBm. FIG. 8 shows the estimated entropy patterns of altered fBm. Note that we did not correct the input signals to ApEn and SampEn by their standard deviation to ensure the same testing condition for all four measures. The results in FIG. 8 show that ApEn and SampEn are more vulnerable to signal amplitude change than RangeEn.

Comment 2: But let's go beyond the in-built normalization invariance.  I would recommend trying to relate the Hurst exponent and RangeEns for a fractional Levy process (as one can vary the parameters of the integrating Levy process while retaining the Hurst exponent).  If you can show a one-to-one relationship between f_m(r) and H for those processes as you sweep over all possible parameters, I'll believe your claim.

 Response and action: Thank you for this useful suggestion. We repeated the same analysis of fBm for a group of fractional Levy motions (fLm’s) at the Levy parameter of alpha=1.25 and varying Hurst exponents from 0.01 to 0.99. The mathematical description of fLm has been added to the manuscript in section 2.2.1 (pages 4-5, lines 89-95). The newly added section 2.3.3. (pages 7-8, lines 164-184) explains the tests that we performed on both fBm and fLm to quantify the relationship between signal entropy measures and the Hurst exponent. Note that in the previous version of the manuscript we did not use signal amplitude correction for estimation of ApEn and SampEn. In the revised manuscript, we use signal amplitude correction and show that SampEn is also related to the Hurst exponent, provided that the tolerance parameter r is multiplied by the input signal’s standard deviation (or equivalently, amplitude of the input signal is divided by the standard deviation).

2.3.3. Signal entropy measures and the Hurst exponent

We also tested the behavior of ApEn, SampEn and RangeEn in presence of self-similarity in random processes. To this end, we simulated fBm and fLm signals with pre-defined Hurst exponents ranging from 0.01 (minimal self-similarity) to 0.99 (maximal self-similarity) with the increasing step of DH=0.01. For each time series, we performed two tests:

 (A) We set a fixed embedding dimension parameter m to 2 (a widely used value across the literature) and computed different entropies over a span of tolerance values r from 0.01 to 1 with 0.01 increasing steps. In this way, we investigated the relationship between a systematic increase in self-similarity (modelled by the Hurst exponent) and the tolerance parameter r in the measures. For each r-trajectory, we estimated the slope of a fitted line to the entropy measures with respect to log(r) and called this quantity r-exponent.

 (B) We set a fixed tolerance parameter r to 0.2 (a common choice in previous studies) and investigated entropy measures over different embedding dimensions m from 2 to 10. Therefore, we examined the relationship between a systematic change in the Hurst exponent and the embedding dimension m. For each m-trajectory, we estimated the slope of a fitted line to the entropy measures with respect to log(m) and called this quantity m-exponent.

We did not perform line fitting for those time series whose extracted entropy measures were undefined for at least one r or m values. We repeated the above tests with and without amplitude correction (i.e., dividing the signal amplitude by its standard deviation). This manual correction step is recommended for ApEn and SampEn analysis, as it can reduce their sensitivity to differences in signal amplitudes (see [24]).

Section 3.4 describes the results of these analyses (page 10, lines 249-279):

 3.4. Relationship with the Hurst exponent

 The analysis results of ApEn and SampEn for fLm signals with different Hurst exponents are summarized in FIG. 4. As seen in FIG. 4 A to D, ApEn and SampEn show a systematic relationship with the Hurst exponent. In particular, SampEn has an inverse monotonic relationship with the Hurst exponent in the r-plane (note the descending color scale along the y-axis at all r values). Although the relationship between ApEn and Hurst is not as monotonic as that of SampEn, it still shows a systematic change. One way of quantifying these changes is by examining their corresponding m-exponents and r-exponents (i.e., the linear slopes of entropy patterns versus log(m) and log(r), respectively). FIG. 4-E to H suggest m- and r-exponents of both ApEn and SampEn are linked to the Hurst exponent in a nonlinear way, before signal amplitude correction. Also, their trajectories reach a plateau in the r and m domains at high Hurst exponent values (note the relatively flat regions of red dots in FIG. 4-E to H). It implies the insensitivity of ApEn and SampEn to high levels of self-similarity, meaning that these measures stay unchanged when the Hurst exponent approaches to 1. Black dotted plots in FIG. 4-E to H suggest that signal amplitude correction results in a more linear relationship between the Hurst exponent and SampEn/ApEn exponents in the r-domain, but it is less pronounced in the m-domain.

 We repeated the same analysis for fBm signals at the maximum range of Hurst exponents (0 < H < 1). The results are illustrated in FIG. 5. As the figure shows, signal amplitude correction has a more significant impact on entropy patterns of fBm than fLm. For example, there is no strong relationship between SampEn and the Hurst exponent without amplitude correction (see FIG. 5-B versus FIG. 5-D). Also, the number of defined SampEn is reduced if we do not perform this correction (note very little number of red dots in FIG. 5-F and no red dot in FIG. 5-H). In addition, similar to the fLm results in FIG. 4, amplitude correction can linearize the relationship between ApEn/SampEn and the Hurst exponent.

 A similar analysis using RangeEnA and RangeEnB highlights their properties in contrast to ApEn and SampEn. The results extracted from fLm and fBm signals are summarized in FIG. 6 and 7, respectively. Firstly, RangeEn trajectories always end at the r value of 0. Secondly, the patterns of RangeEnA and RangeEnB are relavitely similar to each other, except that RangeEnA has a considerable amount of missing (undefined) values, specially over low r values. Thirdly, both RangeEnA and RangeEnB have a more linear relationship with the Hurst exponent than ApEn and SampEn, in r and m trajectories. In particular, the flat regions of their exponents over high H values are shorter than those of ApEn and SampEn (see panels E to H of FIGs. 6 and 7).

Comment 3: Eq. 16: I think it'd be good to point out ways in which this is intuitively not a distance.  (You point out ways in which it is formally not a distance.)  The d_range need not increase as X and X' get farther away from one another.  For instance, take some X and increase all of its component's distances from X' by some epsilon.  Then the numerator of d_range is equivalent, while the denominator increases, and so d_range decreases, even though most would say that X and X' are now farther apart.

 Response and action: We have now added an intuitive description of the range distance in section 2.3.1 as follows (page 6, lines 144-148):

In fact, $d_{range}(\mathbf{X}_i^m,\mathbf{X}_j^m)$ does not necessarily increase as $\mathbf{X}_i^m$ and $\mathbf{X}_j^m$ get farther away from one another by other definitions of distance functions. For instance, assume all elements of $\mathbf{X}_i^m$ are increased by a constant positive value. Then, the numerator of $d_{range}$ stays unchanged, while the denominator increases leading to a reduction in $d_{range}$, even though the Euclidean distance between $\mathbf{X}_i^m$ and $\mathbf{X}_j^m$ has increased.

Comment 4: There are many entropy-based measures that you did not mention that one could use to characterize a time series, though these measures are not as well-studied as those you did mention.  See "Anatomy of a Bit" by R. G. James et al.

 Response and action: Thank you. This reference was added to the Introduction section (reference 26 – pages 1-2, lines 31-41):

A complementary approach to time-frequency distributions and wavelet transforms is signal entropy analysis, a family of methods rooted in information theory. Entropy rate of a random process is defined as the average rate of generation of new information [17]. In this context, independent and identically distributed white noise is assumed to be the projection of thermodynamic equilibrium state into a signal with maximal entropy. This is because identically distributed white noise has a normal distribution where each repeating time point contains new information. On the other hand, a completely regular signal with a repeating pattern of constant values will lead to minimal entropy, as there is no generation of new information. The most prominent types of signal entropy measures may include Shannon entropy [17], Renyi entropy [18], Kolmogorov entropy [19,20], Kolmogorov-Sinai entropy [21], Eckmann-Ruelle entropy [22], approximate entropy (ApEn) [23], sample entropy (SampEn) [24] and multi-scale entropy [25]. See [26] for more examples of entropy-based signal measures.

Comment 5: For the Entropy analysis of EEG data, it seems to me that it might be easy to find something that classifies the time series into healthy, interictal, or ictal.  To that end, I would ask that you calculate the Hurst exponent of the EEG data and see if it clusters; and more than that, also calculate ApEn and sample entropy of the EEG data to see if it clusters by those measures too.

 Response and action: Please note that characterization of ictal periods in epileptic EEG in contrast to interictal intervals is not a trivial challenge in the field of EEG signal processing (see for example the below references).

 [1]          Y. Li, X. Wang, M. Luo, K. Li, X. Yang, and Q. Guo, “Epileptic Seizure Classification of EEGs Using Time–Frequency Analysis Based Multiscale Radial Basis Functions,” IEEE J. Biomed. Health Inform., vol. 22, no. 2, pp. 386–397, Mar. 2018.

[2]          M. Nova, L. Vyslouzilova, Z. Vojtech, and O. Stepankova, “Towards Computer Supported Search for Semiological Features in Epilepsy Seizure Classification,” in World Congress on Medical Physics and Biomedical Engineering 2018, 2019, pp. 363–366.

 In order to make our EEG analysis more biologically meaningful, we have now excluded scalp-level EEG datasets (i.e., datasets A and B) from our study and limit it to the epileptic intracranial recordings at two states: seizure (ictal) and non-seizure (interictal) states. We then performed your suggested Hurst exponent extraction and also, did the analysis for all four entropy measures (in contrast to RangeEn analysis only in the previous version of our manuscript). Section 3.6 and FIG. 9 have now been dedicated to this (pages 10-11, lines 290-298):

3.6. Analysis of epileptic EEG

We performed self-similarity analysis of epileptic EEG datasets by extracting their Hurst exponent through the standard rescaled range approach [41]. FIG. 9-A illustrates the distributions of Hurst exponents for three datasets. Whilst interictal segments are clustered towards higher self-similarity levels, ictal segments have been distributed across a wider range between high and low self-similarity. FIG. 9-B to E represent the patterns of ApEn, SampEn, RangeEnA and RangeEnB in the r-plane for the three EEG datasets (corrected amplitudes and fixed m of 2). In all plots, the two interictal r-trajectories are close to each other and represent a relatively different trajectory to the ictal state. However, this distinction is more apparent for the RangeEn measures.

Accordingly, the below paragraph was updated in the Discussion section (page 12, lines 360-374):

A realization of brain complexity is reflected in EEG signals which convey information about electrical activity of neuronal populations within cortical and sub-cortical structures. Epilepsy research is a field that significantly benefited from EEG analysis, as the disease is associated with abnormal patterns in EEG such as seizures and interictal epileptiform discharges [43]. Therefore, characterization of abnormal events in epileptic EEG recordings is of great significance in diagnosis, prognosis and management of epilepsy [27,44,45]. Our results suggest that interictal EEG at the intracranial level is more self-similar than ictal EEG with clustered Hurst exponents towards 1. On the other hand, the Hurst exponent of ictal EEG covers high and low self-similarity (see FIG. 9-A). Therefore, self-similarity may not be a discriminative feature of ictal state. On the other hand, all entropy measures represent distinctive trajectories in the r-plane for interictal versus ictal states with relatively low variance over EEG segments (see FIG. 9-B to E). This implies that signal complexity analysis of entropy may be more beneficial than self-similarity analysis for epileptic seizure detection and classification. Note that in the absence of any time-varying artefact, EEG signals can be considered as weak stationary processes [49]. Therefore, a correction of the amplitude changes by standard deviation of the signal in ApEn and SampEn probably may lead to comparable results with RangeEn.

Comment 6: Line 27: I'd say the entropy rate, rather than the entropy.  It would be a shame if the first sentence were to confuse the reader between the single-symbol entropy and the entropy rate.

 Response and action: Thank you for your correction, but we have now excluded the sections related to Shannon, Kolmogorov-Sinai and Eckmann-Ruelle entropies as per reviewer four’s suggestion and also, their irrelevance to the updated study.

Comment 7: Line 30: It's not a uniform probability density-- it's a normal distribution.

 Response and action: Corrected.

Comment 8: Line 34: Kolmogorov, not Kulmogorov.  Also, at the end of this paragraph, it might be worth mentioning that in Section 2, you will explain how all of these relate.

 Response and action: Please see our response to your Comment 6.  

Comment 9: Line 52: Not a fan of the phrase "have better consistency", since I don't think it's clear what you mean.  Do you mean that N need not be as large in order to be close to the N-->infinity answer?

 Response and action: We have re-written the relevant paragraph in the Introduction section as follows (page 2, lines 54-59):

 In this study, we investigate the behaviour of ApEn and SampEn in presence of self-similarity and examine their relationship with the Hurst exponent through their tolerance and embedding dimension parameters. We also address the issue of sensitivity to signal amplitude changes in ApEn and SampEn and develop modified versions called range entropies or RangeEn. We compare RangeEn with ApEn and SampEn from different perspectives using multiple simulations. Finally, we demonstrate the capacity of signal entropy measures for epileptic EEG characterization.

Comment 10: Eq. 2 gives the single-symbol entropy, whereas I think you want the formula for the entropy rate, which requires conditioning on pasts of symbols.

 Response and action: In the updated version, Eq. 2 has been excluded. Please see our response to your Comment 6.

Comment 11: In Line 67, what do you mean by "can be quantified by K in Eq. 2"?  The K is merely a constant to go between different logarithmic bases.

 Response and action: The section related to the Kolmogorov-Sinai entropy has been excluded in the updated version. Please see our response to your Comment 6.

Comment 12: Line 68: Might want to mention that Pesin's theorem is useful only for deterministic dynamical systems.  As a result, I also think that here you should discuss the epsilon-tau entropy rate and give the definition of the K-S entropy (rate) in terms of the epsilon, tau entropy rate, and then cite Pesin's theorem and give some of its conditions so that its limitations are clear.

 Response and action: The section related to the Kolmogorov-Sinai entropy has been excluded in the updated version. Please see our response to your Comment 6.

Comment 13: Line 75: Again, do you mean single-symbol entropy or entropy rate?

 Response and action: Again, this sentence has been excluded from the updated version.

Comment 14: Line after 76: "spacial" should be special

 Response and action: Corrected. Thank you.

Comment 15: First paragraph of Section 3: "behviour" should be "behavior" and "centred" should be "centered"

 Response and action: These terms have been corrected throughout the manuscript.

Round  2

Reviewer 4 Report

Thank you for performing the suggested analyses!